# Yield Surfaces and Plastic Potentials for Metals, with Analysis of Plastic Dilatation and Strength Asymmetry in BCC Crystals

**Aleksander Zubelewicz [1],\* and John D. Clayton [2]**

1   Alek and Research Associates, Los Alamos, NM 87544, USA
2   Army Research Laboratory, Aberdeen, MD 21005, USA
\*   Correspondence: alek.zubelewicz@gmail.com

**Abstract:** Since the 1980s, constitutive modeling has steadily migrated from phenomenological descriptions toward approaches that are based on micromechanics considerations. Despite significant efforts, crystal plasticity remains an open field of research. Among the unresolved issues are the anomalous behavior of metals at low temperatures and the stress upturn at extreme dynamics. This work is focused on the low-temperature responses of body-centered-cubic (bcc) metals, among them, molybdenum (Mo). At these conditions, the plastic flow strength is governed by the motion of screw dislocations. The resultant non-planarity of core structures and slip causes the following: the shear stress includes non-glide components, the Schmid law is violated, there is a tension-compression asymmetry, and the yield surface and plastic potential are clearly decoupled. We find that the behavioral complexities can be explained by atomistically resolved friction coefficients in macroscopic yield and flow. The plastic flow mechanisms establish the departure point into the follow-up analysis of yield surfaces. For example, we know that while the von Mises stress is explained based on energy considerations, we will also show that the stress has a clear geometric interpretation. Moreover, the von Mises stress is just one case within a much broader class of equivalent stresses. Possible correlations among non-Schmid effects (as represented macroscopically by friction coefficients), volume change (i.e., residual elastic dilatation) from dislocation lines, and elastic anisotropy are investigated. Extensions to the shock regime are also established.

**Keywords:** tensor representations; friction coefficient; metal plasticity; dislocations; Schmid law

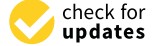



## 1. Introduction

The mechanisms of plastic deformation in bcc metals at low and medium temperatures are different from the observed mechanisms in face-centered cubic (fcc) and hexagonal-close-packed (hcp) metals. The plastic flow is controlled by the motion of 1/2<111> screw dislocations. The dislocations may spread out into several planes of the <111> zone [1,2]. In single crystals, the plastic flow exhibits anisotropic characteristics, and there is a tension-compression asymmetry. The non-planar structure of the dislocation cores is responsible for high friction stress (i.e., Peierls stress), there is an asymmetry of the yield stress in tension and compression, the Schmid law is violated, and the yield surface and plastic potential are clearly decoupled [3–6]. Consequently, a work-conjugate pair of the equivalent stress and the rate of plastic strain cannot be constructed. At increasing temperatures, the tension-compression asymmetry is reduced, and this non-Schmid effect nearly vanishes at room temperature [7].

In metal plasticity, most phenomenological constitutive models are formulated in the framework of von Mises ($J_2$) theory. Also well known, but less popular in practice, is Tresca plasticity. The obvious advantage of the Huber–von Mises yield surface is its numerical convenience. The existence of a smooth and convex yield surface makes the analysis numerically friendly. In contrast, the Tresca surface has built-in singularity points/lines, which pose issues when constructing the associated plastic flow rules. We emphasize that

the two approaches were introduced over 100 years ago; the Tresca plasticity was proposed in 1864 [8], the energy-based criterion was suggested in 1904 by Huber [9,10], and the concept of plastic flow was formulated by von Mises in 1913 [11]. Tresca plasticity assumes that the plastic slip is initiated when the maximum shear stress reaches a critical magnitude. Still, we are aware that the active slip planes may deviate from the plane of maximum shear. Such misorientations are quantified by the Schmid factor. According to the Schmid law [12], plastic flow begins when the resolved shear stress on a given slip plane attains the threshold level known as the critical resolved shear stress. This law also implies that the driving force is not influenced by other components of stress. Taylor [13] and many other researchers found that the law is not applicable to bcc metals, as it has limited justification [5].

In contrast to the Tresca concept, Huber–von Mises plasticity is formulated based on energy considerations, where the plastic deformation begins when elastic energy (deviatoric part only) exceeds a certain energy barrier. In the Huber–von Mises concept, slip planes are not defined, and as explained in Section 2, the Schmid law should be used with some caution as well. It is worth mentioning Hosford's yield criterion for isotropic plasticity [14], which allows reshaping the stress envelope. In this and many other phenomenological models, emphasis is placed on the definition of the material's strength (or yield stress), while little attention is paid to the actual mechanisms of plastic flow. Also noteworthy in this context is Hershey's description of isotropic plasticity [15] and preceding seminal treatments of yielding by Taylor, Bishop, and Hill [16–18]. Although the current work focuses on macroscopically isotropic polycrystalline responses for untextured metals, prominent asymmetric yield criteria for materials of lower symmetry (e.g., orthotropy) are also noted, e.g., [19–21].

While recognizing the importance of yield stress, we focus our investigations on the mechanisms of deformation, and then, we determine whether a coupling of the flow mechanisms with the equivalent stresses exists or, just as important, whether the yield surface should be treated independently from the plastic potential. Herein, the analysis is based on the tensor representation method (TRM) developed in [22]. In Section 2, we illustrate TRM capabilities by constructing a geometric interpretation for the Huber–von Mises flow mechanism. Then, in Section 3, we focus the discussion on the flow mechanisms in bcc Mo. In what follows, we can identify two friction coefficients that capture the effects of the slip non-planarity. One of the coefficients characterizes the yield surface, and the second is used in the flow potential. We show that the coefficients properly reproduce the stress asymmetry in molybdenum at low temperatures, and the friction term can depict the change of flow mechanism at shock conditions (Appendix A). In Section 4, possible connections between the macroscopic coefficients, elastic anisotropy, and dislocation core effects are explored theoretically, with a focus on origins of local plastic dilatation.

## 2. Geometric Interpretation of Huber–von Mises Flow Mechanism

We begin by formulating a geometric interpretation for the Huber–von Mises plastic flow. The procedure is an important step because it explains further generalizations of the flow mechanisms and yield surfaces for bcc metals. More specifically, we want to determine the dominant slip planes, which might be associated with the $J_2$ stress envelope. As we know, the von Mises flow tensor $M_{ij} = \frac{\sqrt{3}\, S_{ij}}{\sqrt{J_2}}$ is defined in terms of the stress deviator $S_{ij} = \sigma_{ij} - p\delta_{ij}$, where $\sigma_{ij}$ is the Cauchy stress, while the pressure $p = \sigma_{kk}/3$ (here, defined as positive in tension) and the Kronecker delta $\delta_{ij}$ complete the relation. The flow tensor $M_{ij}$ specifies the mechanism of plastic flow such that $\dot{\varepsilon}_{ij}^p = \frac{1}{2} M_{ij} \dot{e}_{eq}^p$. When the mechanism is coupled with stress $\sigma_{ij}\dot{\varepsilon}_{ij}^p = \left( \frac{1}{2} M_{ij}\sigma_{ij} \right) \dot{e}_{eq}^p$, the equivalent stress becomes $\sigma_{eq} = M_{ij}\sigma_{ij}/2 = \sqrt{3J_2}$; i.e., it is the Huber–von Mises stress. We find that the flow tensor $M_{ij}$ can be expressed in terms of three eigentensors: $N_{ij}^1$, $N_{ij}^2$, and $N_{ij}^3$ such that the tensors are aligned with the principal stresses $\sigma_1 = N_{ij}^1\sigma_{ij}$, $\sigma_2 = N_{ij}^2\sigma_{ij}$, and $\sigma_3 = N_{ij}^3\sigma_{ij}$, where $\sigma_1 \geq \sigma_2 \geq \sigma_3$ and $N_{ij}^1 + N_{ij}^2 + N_{ij}^3 = \delta_{ij}$. Before proceeding, it is important that we introduce

the tensor representation method and show that a generic eigentensor can be uniquely expressed in terms of other second-order tensors, for example, the stress tensor.

*2.1. Tensor Representations*

The procedure for constructing tensor representations $N_{ij}^1$, $N_{ij}^2$, and $N_{ij}^3$ is described by Zubelewicz [22], where detailed derivations can be found. As stated, any symmetric second-order tensor can be represented by another second-order symmetric tensor if the original tensor and its representation produce the same invariants. Here, the generic dyadic product $N_{ij} = n_i n_j$ is constructed on a unit vector $n_k$. It is clear that $N_{kk} = 1$, $N_{ik} N_{ki} = 1$, and $N_{ik} N_{kl} N_{li} = 1$, as is true for the tensor taken to any power. We also know that any second-order symmetric tensor can be expressed in the form of three fundamental terms. Since the plastic flow is controlled by the current stress, the tensor representation of $N_{ij}$ will be defined in terms of stress or, here, with the use of the stress deviator. In fact, the stress and stress-deviator-based representations are equivalent; hence,

$$N_{ij}^m = a_m \delta_{ij} + b_m S_{ij} + c_m S_{ik} S_{kj} \tag{1}$$

The superscript $m$ in $N_{ij}^m$ indicates the direction of the principal stress. As stated earlier, there are three relevant invariants: $N_{kk} = 1$, $N_{ik} N_{ki} = 1$, and $N_{ik} N_{kj} N_{ji} = 1$. The requirement is that the representation (1) retains the same invariants as the generic eigentensors. Consequently, there are three invariants and three conditions, and, upon solving the equations, we have three sets of parameters $\{a_m, b_m, c_m\}$. The first tensor $N_{ij}^1$ reproduces the dyadic product constructed on the unit vector pointing in the direction of the maximum tensile stress. The parameters for $m = 1$ are

$$a_1 = \tfrac{1}{3} - \tfrac{2}{3} \cos \tfrac{\pi+\varphi}{3} \sec \varphi$$

$$b_1 = \frac{\cos\left(\tfrac{\pi}{6} + \tfrac{2\varphi}{3}\right)}{\sqrt{3J_2}} \sec \varphi \tag{2}$$

$$c_1 = \frac{\cos \tfrac{\pi+\varphi}{3}}{J_2} \sec \varphi$$

The tensor $N_{ij}^2$ is a dyadic product associated with the second principal stress. In this case, the parameters for $m = 2$ are

$$a_2 = \tfrac{1}{3} + \tfrac{2}{3} \cos \tfrac{\varphi}{3} \sec \varphi$$

$$b_2 = \frac{\sin \tfrac{2\varphi}{3}}{\sqrt{3J_2}} \sec \varphi \tag{3}$$

$$c_2 = - \frac{\cos \tfrac{\varphi}{3}}{J_2} \sec \varphi$$

Lastly, the third tensor $N_{ij}^3$ determines the orientation of maximum compression, where for $m = 3$,

$$a_3 = \tfrac{1}{3} - \tfrac{2}{3} \cos \tfrac{\pi-\varphi}{3} \sec \varphi$$

$$b_3 = - \frac{\cos\left(\tfrac{\pi}{6} - \tfrac{2\varphi}{3}\right)}{\sqrt{3J_2}} \sec \varphi \tag{4}$$

$$c_3 = \frac{\cos \tfrac{\pi-\varphi}{3}}{J_2} \sec \varphi$$

In this construction, the second and third invariants of the stress deviator (i.e., $J_2 = S_{ij}S_{ij}/2$ and $J_3 = S_{ik}S_{kj}S_{ji}/3$) define the angle $\varphi = \sin^{-1}(A_\varphi)$, where $A_\varphi = 3\sqrt{3}J_3 / \left(2J_2^{3/2}\right)$.

The angle $\varphi$ varies between $\pm\pi/2$. Once again, the stress representations of the eigentensors must satisfy the condition $N_{ij}^1(S_{kl}) + N_{ij}^2(S_{kl}) + N_{ij}^3(S_{kl}) = \delta_{ij}$. In short, the

TRM is a very useful tool for researchers. It has already been shown that experimentally observed flow mechanisms, at first constructed in a generic tensorial form, can be uniquely coupled with the driving tensorial stimuli [23]. In this manner, we eliminate the uncertainty about the functional form of the mechanisms.

### 2.2. Atomistically Resolved Friction Coefficient

Once again, here we attempt to construct a geometric interpretation for the flow mechanism $M_{ij} = \frac{\sqrt{3}\, S_{ij}}{\sqrt{J_2}}$. In the first step, we construct a generalized slip mechanism along three planes, where the planes are corotational with principal stresses such that

$$M_{ij} = \alpha \left( N_{ij}^1 - N_{ij}^3 \right) + \beta \left( N_{ij}^1 - N_{ij}^2 \right) - \beta \left( N_{ij}^2 - N_{ij}^3 \right) \tag{5}$$

Note that each plane is weighted by functions $\alpha$ and $\beta$. Next, we recall the definition of the flow tensor $M_{ij} = \frac{\sqrt{3}\, S_{ij}}{\sqrt{J_2}}$, where the scalar products are equal to $M_{kk} = 0$, $M_{ij}M_{ij} = 6$ and $M_{ik}M_{kj}M_{ji} = 6 \sin\varphi$, respectively. From there, we identify the two functions $\alpha$ and $\beta$. The functions take the following form:

$$\alpha = \sqrt{3}\, \cos\frac{\varphi}{3}$$

$$\beta = \beta_0 \sin\frac{\varphi}{3} \tag{6}$$

In the next step, the expression (5) is reorganized and presented in an equivalent form $M_{ij} = \alpha \left( N_{ij}^1 - N_{ij}^3 \right) + \beta \left( N_{ij}^1 + N_{ij}^2 + N_{ij}^3 - 3N_{ij}^2 \right)$, and then, the expression is presented in the final form

$$M_{ij} = \alpha \left[ \left( N_{ij}^1 - N_{ij}^3 \right) + \mu_\varphi \left( \delta_{ij}/3 - N_{ij}^2 \right) \right] \tag{7}$$

Note that $N_{ij}^1 + N_{ij}^2 + N_{ij}^3 = \delta_{ij}$. The parameter $\mu_\varphi = 3\beta/\alpha$ is interpreted as an atomistically resolved friction coefficient. The function $\alpha$ varies between $3/2$ and $\sqrt{3}$; thus, the function is nearly a constant. The friction coefficient $\mu_\varphi$ takes values between $\mp\beta_0$. At first glance, the relation resembles the Coulomb law used in frictional materials [24]. A generic form of the flow tensor, but not a stress representation, was introduced in [25]. In the current application to bcc metals, we realize that the coefficient

$$\mu_\varphi = \sqrt{3}\, \beta_0 \, \tan\varphi/3 \tag{8}$$

quantifies the slip non-planarity [1,5,26,27]. Consequently, the Huber–von Mises stress $\sigma_{eq} = \sqrt{3J_2}$ is equal to

$$\sigma_{eq} = \frac{\alpha}{2} \left[ (\sigma_1 - \sigma_3) + \mu_\varphi (p - \sigma_2) \right] \tag{9}$$

In the flow tensor (7) and in the equivalent stress (9), the slip non-planarity is quantified in the second terms $\mu_\varphi \left( \delta_{ij}/3 - N_{ij}^2 \right)$ and $\mu_\varphi (p - \sigma_2)$. In frictional materials, the second term characterizes the roughness of the slip surface, where the roughness is responsible for dilatational inelastic deformation.

In metals, too, the atomistically resolved friction coefficient $\mu_\varphi$—here, directly proportional to $\beta_0$—mediates the effect of non-glide parts of the stress. However, the non-planarity does not affect the material's volume in a meaningful manner, at least at the continuum (macroscopic) scale. Therefore, plasticity is essentially volume-preserving, in an average sense, when the local volume element contains a dislocation density not exceeding $10^{15}/\text{m}^2$. In Appendix A, we show that the isochoric flow assumption cannot be over-generalized to shock-loading regimes, with higher dislocation densities, rising adiabatic temperature, and local excitations; all the factors magnify core pressure.

In summary, the Huber–von Mises stress is preserved when the parameter is equal to unity, i.e., $\beta_0 = 1$, as shown by the black line in Figure 1. However, several other surfaces

can be constructed. A Tresca-like criterion (blue line) is obtained for $\beta_0 = 1/2$. The true Tresca stress envelope (red line) is constructed by prescribing $\alpha = 2$ and $\beta_0 = 0$. In the Tresca-like criterion, singularity points are rounded with $A_\varphi = 3\sqrt{3}\,(1 - A_0)\,J_3 / \left(2 J_2^{3/2}\right)$, where $A_0 = 0.2$. Herein, the equivalent shear stress and the rate of plastic strain represent the work-conjugate pairs, that is, $\sigma_{ij}\dot{\varepsilon}_{ij}^p = \left(\frac{1}{2}\,M_{ij}\sigma_{ij}\right)\dot{e}_{eq}^p$ and $\sigma_{eq} = M_{ij}\sigma_{ij}/2$. It is worth noting that the friction parameter $\beta_0$ in (8) makes our stress envelopes (9) comparable to the Hershey–Hosford criteria for fcc polycrystals [14] with a large exponent.

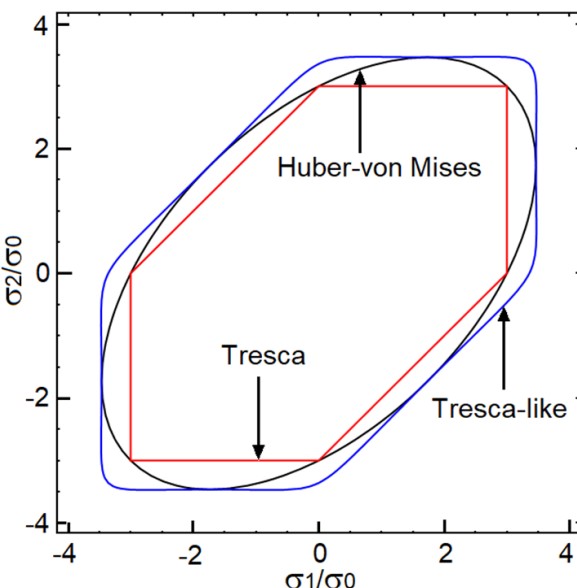

**Figure 1.** Yield stress plotted on the stress plane $(\sigma_1, \sigma_2)$, where $\sigma_3 = 0$. The original Tresca criterion (red line) is obtained for $\alpha = 2$ and $\beta_0 = 0$. The Huber–von Mises criterion (black line) is found from setting $\beta_0 = 1$. In the Tresca-like yield criterion $\beta_0 = 1/2$, and the line is colored blue.

## 3. Generalized Huber–von Mises Criterion

In fcc polycrystalline metals, slip is activated along planes somewhat misoriented with respect to the plane of maximum shear, where the Schmid factor properly quantifies the degree of misorientation. However, in bcc metals, the breakdown of the Schmid law is a known fact, as reported in many studies, e.g., [5,28,29]. Specifically, at low temperatures, plastic flow depends on the resolved shear stress and is also affected by other stress components. The non-planarity of the dislocation core structure is the main reason for the observed strength asymmetry in tension and compression. As reported in [1,26,27], the yield surface and the plastic potential are decoupled. Molecular dynamics (MD) calculations for Mo crystals support the construction of yield surfaces and plastic potentials [26,27]. A prior conclusion from [1,5,27] was that the work-conjugate pair of stress and plastic strain rate cannot be established in bcc polycrystals at low temperatures.

In bcc metals, the non-planarity of the dislocation core arises at the atomistic scale and affects the shear stress (i.e., screw dislocation core spreading onto multiple planes, with possible edge components within the core structure [5]); still, still the plastic flow remains nearly incompressible for dislocation densities far below the theoretical maximum limit (see Section 4). The friction coefficient denoted by $\mu_0$ quantifies the core non-planarity such that

$$\mu_\varphi = \sqrt{3}\,\beta_0\,\tan\varphi/3 - \mu_0 \tag{10}$$

Again, in reference to frictional materials, the internal friction angle $\varphi/3$ characterizes the angle of asperities. This angle changes and is a function of the current stress or, more precisely, the direction of the maximum shear stress. Thus, the friction mechanism in metals is endowed with much higher configurational flexibility. We reemphasize that the

internal friction parameter (8) replicates the relation used in the Coulomb law. In the case of molybdenum, studies in [27] indicate that $\beta_0 = 2/3$ properly captures the shape of the stress envelope, as seen in Figure 2. One should note that the parameter $\beta_0$ scales the non-planarity of the plastic flow. For example, in fcc metals, the parameter should be equal to $\beta_0 = 1/2$, while $\beta_0 = 1$ depicts the strongest out-of-plane contribution. In this construction, the difference between the yield stress and the plastic potential is solely controlled by the parameter $\mu_0$. The yield stress becomes

$$\sigma_Y = \frac{\alpha}{2} \left[ (\sigma_1 - \sigma_3) + \mu_\varphi^Y (p - \sigma_2) \right] \tag{11}$$

where the non-planarity coefficient entering Equation (11) is found to be $\mu_0^Y = 0.3$ for Mo. The strength differential $SD_Y = 2(\sigma_t - \sigma_c)/(\sigma_t + \sigma_c)$ introduced by Vitek et al. [1] characterizes the tension–compression asymmetry, where $\sigma_c$ and $\sigma_t$ are uniaxial stresses in compression and tension. Here, the differential takes the very simple form $SD_Y = \beta_0 \, \mu_0^Y$. A similar differential is calculated for the plastic potential: $SD_P = \beta_0 \, \mu_0^P$, where the superscript "$P$" is added to (10). Now, we have

$$\mu_\varphi^P = 2/\sqrt{3} \, \tan \varphi / 3 - \mu_0^P \tag{12}$$

where $\mu_0^P = -0.09$ for Mo. The flow mechanism becomes

$$M_{ij}^P = \alpha \left[ \left( N_{ij}^1 - N_{ij}^3 \right) + \mu_\varphi^P \left( \delta_{ij}/3 - N_{ij}^2 \right) \right] \tag{13}$$

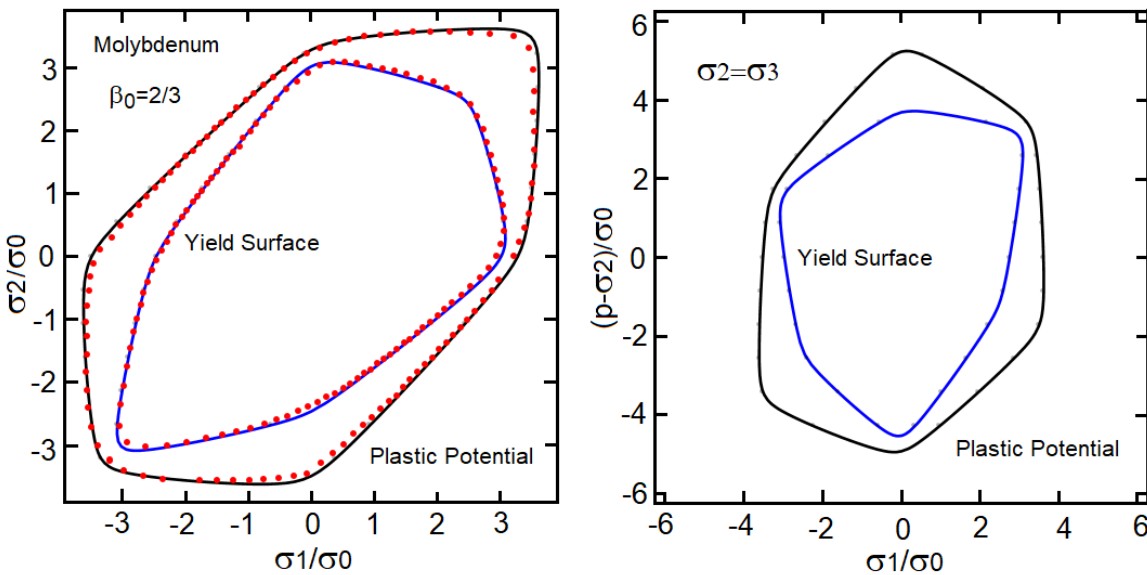

**Figure 2.** Yield surface (black line) and plastic potential (blue line) plotted on the plane of principal stresses $\sigma_1$ and $\sigma_2$. Red data points are based on polycrystal model predictions with slip strengths informed by MD studies of glide of screw dislocations in bcc Mo, simulated by Gröger et al. [26,27]. The original source of the red data points is ref. [27]; the blue and black lines are the new output of the model set forth in Section 3 of the current work. The right-hand-side plot depicts the surface's asymmetry on the plane of $\sigma_1$ and $(p - \sigma_2)$, where $p$ is hydrostatic pressure.

In the absence of damage or point defects (e.g., no vacancies, interstitials, or inclusions), and at dislocation densities sufficiently low, Mo can be modeled as a plastically incompressible material: $M_{kk}^P = 0$. The plastic potential $\sigma_P = \frac{1}{2} M_{ij}^P \sigma_{ij}$ takes the following form:

$$\sigma_P = \frac{\alpha}{2} \left[ (\sigma_1 - \sigma_3) + \mu_\varphi^P (p - \sigma_2) \right] \tag{14}$$

The isotropic yield surface and the plastic potential for polycrystalline Mo at low temperatures are shown in Figure 2. Red data points represent the yield and flow surfaces for Mo polycrystals, as in Figure 6 of Gröger et al. [27]. The latter are obtained from Taylor-type [16] crystal plasticity calculations on randomly oriented aggregates of Mo polycrystals [26,27]. For clarity, we used the data reported in [27] in our previous study [25] as well, but now the material model has been recalibrated and the yield surface and plastic potential are redrawn accordingly. Note that the rate of plastic work is equal to $\sigma_{ij}\dot{\varepsilon}^p_{ij} = \frac{1}{2}\left[\left(M^P_{ij} - M^Y_{ij}\right) + M^Y_{ij}\right]\sigma_{ij}\dot{e}^p_{eq}$. Consequently, the plastic power becomes

$$\sigma_{ij}\dot{\varepsilon}^p_{ij} = \left[\sigma^Y_{eq} + \left(\mu^Y_0 - \mu^P_0\right)(p - \sigma_2)\right]\dot{e}^p_{eq} \tag{15}$$

The associated plastic flow is reestablished when $\mu^Y_0 = \mu^P_0$.

The right-hand-side plot in Figure 2 displays strong influence of the "friction" term on the plastic potential (14) and yield stress (11). The stress envelopes are clearly asymmetric. The tension–compression asymmetry is a well-established fact [1,5,26,27]. The dominant tensile loading intensifies the plastic flow, and the opposite is true for compression. The single-crystal slip system strengths of Gröger et al. [26,27] properly capture experimental trends on tension–compression asymmetries observed in experiments on Mo [30–32] at a temperature of 123 K. The current model nearly perfectly matches the yield and flow surfaces of Figure 6 in ref. [27] using just two parameters, $\beta_0$ and $\mu_0$, where each of them has a well-defined physics interpretation; the agreement rationalizes the mechanisms-based constitutive description. Note that Figure 2 of the current work contains solid black and blue curves that are generated as the output of our new model. Figure 2 is an original figure created by the present authors; it is not a scanned reproduction of Figure 6 of ref. [27], which contains differently shaped curves from a different model.

## 4. Screw Dislocations in bcc Metals: Core Spreading and Volume Changes

The forthcoming analysis serves two major purposes. Firstly, the hypothesis that a correlation exists between local dilatation from dislocation lines with dislocation core spreading is examined through theoretical calculations. By the inverse argument, if all atomic motion were restricted to a single plane (i.e., no core spreading onto multiple planes), then the lattice distortion should consist only of simple shearing modes, and no volume change should occur. If a positive/dilatative volume change does occur, then external compressive pressure would work negatively against such a change at very small scales, which could induce extra glide resistance manifesting as non-Schmid effects. Possible correlations with elastic anisotropy are also newly investigated.

Secondly, the analysis theoretically predicts the maximum volume changes expected from dislocation lines for bcc metals (Mo, W, and Ta) to evaluate the proper domain of plastic incompressibility assumed in Sections 2 and 3. Calculations have been reported previously for select fcc metals and Fe [33–35] but not for Group VIB metals which show strong dislocation core spreading and non-Schmid effects and thus potentially more plastic dilatation. Backgrounds on bcc screw dislocation physics and analytical models are given in Sections 4.1 and 4.2 to set the context; new contributions follow in Section 4.3.

### 4.1. Background: Dislocation Core Phenomena

In bcc metals, screw dislocation mobility is generally much lower than edge dislocation mobility. Thus, the yield and flow of bcc metals are dominated by the glide resistance of screw dislocation components, which becomes the limiting factor regarding plastic strength [36,37]. Primary slip systems are <111>{110} and <111>{112}; these are the glide systems typically studied in MD investigations [5,38] and resolved in continuum crystal plasticity models of bcc metals, for example [39].

As reviewed by Duesbery and Vitek [5,38], non-Schmid effects in bcc crystals arise from two primary factors. The first is solely due to the lack of certain symmetry in the bcc crystal structure: strengths may differ when slip occurs in the twinned or anti-twinned

oriented <111>{112} systems. The second is due to the unusual core structures of screw dislocations in bcc metals. The dislocation core tends to spread onto multiple planes, rather than being confined to a single {110} plane, for example. Within the core, the <111> screw dislocation (when viewed macroscopically) contains atomic-scale perturbations of both edge and screw character [5,38]. The fractional Burgers vector components of edge character in the core must sum to zero, such that the <111> dislocation remains of pure screw character macroscopically. The partial edge components contribute strongly to observed non-Schmid effects on yield and flow stresses. For shuffling and glide of the <111> screw dislocation to occur, the non-planar components must first be forced to return to a single dominant glide plane; i.e., the core spreading must be compacted. This can contribute to a rather large Peierls stress in bcc metals. In the macroscopic continuum theory of Section 3, friction coefficients depict an excessive spreading of <111> screw dislocations.

The effects of core spreading (i.e., non-planarity) are more prevalent at very low temperatures (e.g., far below room temperature, typically at or near 0 K in atomic simulations and 77–123 K in experiments [5,27,30–32]); the present discussion is focused on isothermal behavior at low temperature limits. Thermal activation, thermal expansion, and phonon drag are of no great relevance here but are applicable to shock conditions in Appendix A.

Observed non-Schmid effects vary in magnitude among bcc metals, being stronger in Group VIB crystals (e.g., Mo and W) than in Group VB crystals (e.g., Ta). Perfect crystals of the Group VIB metals have larger elastic constants than those in Group VB, which, as noted by Duesbery and Vitek [5], could exacerbate their core spreading. The above trends were deduced primarily from investigations [5,36,38] that employed empirical interatomic potentials for the behavior of bcc metals, for example, the Finnis-Sinclair potential [40]. More contemporary MD potentials [1,26,27,38] and first-principles methods (i.e., density functional theory (DFT)) such as tight binding [41] have confirmed the existence of the coupled phenomena of core spreading, non-planarity, and non-Schmid effects; details of core structures and stress differentials for asymmetric slip can differ among models [38].

### 4.2. Background: Volume Changes from Dislocations

The isotropic linear elastic solution for a screw dislocation [42,43] predicts that no local or global volume change manifests from its elastic fields. For edge dislocations, according to the isotropic linear elastic solution, local volume change occurs in the vicinity of the dislocation line, but no global volume change (or global shape change, for that matter) can occur for an externally unloaded (i.e., traction-free or self-equilibrated) body containing edge (or screw) dislocations in the context of pure *linear* elasticity theory, isotropic or anisotropic [44,45]. This statement is strictly true for isothermal conditions, wherein no thermal expansion or contraction from atomic vibrations in the vicinity of defects occurs.

Volume changes from glide dislocations (e.g., no vacancies from dislocation climb, and no other point defects) can arise from two notable sources in an isothermal continuum elasticity theory: nonlinear elastic effects (i.e., isothermal anharmonic effects) and dislocation core pressure. The latter can be modeled, in the context of a cylindrical annulus of elastic material enclosing a straight dislocation line, by a pressure boundary condition acting on the inner surface of the annulus, within which the core resides [46]. Atomic calculations can estimate its magnitude, for example, up to the order of 10% of the shear or bulk modulus [46,47]. However, core pressure varies inversely with the squared radial distance from the defect line [48], so its magnitude depends on the choice of core radius.

Anharmonic effects, which could induce residual lattice shape change as proven by Clayton and Bammann [43,45] in addition to volume change, can be associated with the nonlinear elastic constitutive response of the crystal, through a combination of second- and third-order elastic constants in crystals of arbitrary symmetry. For crystals of cubic symmetry, the volume change component of average residual deformation depends on the combination of second- and third-order elastic constants and particular parts of the elastic energy density [34,43–45]. Although partitioning of the elastic energy density

into the requisite components does not seem readily available for known anisotropic solutions [49,50], this partitioning may be analytically possible.

On the other hand, closed-form solutions for volume change due to dislocation lines are readily derived for the isotropic case [51] considering anharmonic terms of third order in the strain energy density, when the dislocation line energy is estimated from linear isotropic elasticity with a suitable cutoff and core radius. In the isotropic case, the requisite second- and third-order elasticity coefficients are reduced to the usual two linear (i.e., second-order) elastic constants and the ambient pressure derivatives of the shear and bulk moduli. As shown later, the normalized effect of second-order constants can be resolved solely by Poisson's ratio. Different derivations based on nonlinear continuum mechanics [43–45] or thermodynamic arguments [33,52] can arrive at similar end results.

Dilatation from edge and screw dislocations in bcc α-iron has been confirmed using MD potentials and DFT [53–55]. As noted by Clouet et al. [54,55], dilatation from <111> screw dislocation lines can also be deduced from DFT results for pure Mo [23].

Trends, e.g., expansion rather than contraction predicted for engineering metals, and of fairly small magnitude, agree with limited experimental data [33–35,43,45], as will be shown in Section 4.3. Unless the local dislocation density is extremely large, the net residual volume change from dislocations is generally considered small enough to be ignored in continuum plasticity theory for standard, as opposed to extreme, loading conditions. Yield and flow stresses can still depend on pressure even when plastic volume change is omitted in kinematics, as in Section 3 herein (but not Appendix A) or in other models [35].

### 4.3. Analysis: Volume Changes in bcc, fcc, and hcp Metals

A question to be investigated next is whether non-Schmid effects leading to nonzero lattice friction coefficients in the continuum plasticity theory of Section 3 correlate with dilatation from dislocation lines due to anharmonicity or whether any such correlation manifests only from an independent core pressure. Contributions of anharmonicity and core pressure to plastic volume change are expressed as follows in the isotropic elastic limit, consolidating prior nonlinear elastic and atomic-scale derivations [43] (Ch. 7), [45].

Denote the total density of dislocation lines, in dimensions of length per unit reference volume, as $\rho$, with $b$ the magnitude of the full Burgers vector. Denote $f^e = \chi$ as the fraction of this density of edge character, and $f^s = 1 - \chi$ the fraction of screw character, whereby definition $f^e + f^s = 1$. Let $B$ and $G$ label the ambient bulk modulus and ambient shear modulus, and $B' = \mathrm{d}B/\mathrm{d}p$ and $G' = \mathrm{d}G/\mathrm{d}p$ the derivatives of bulk and shear moduli with respect to the external pressure $p$ (here, the usual convention is positive in compression) measured in the reference state. Let $p_c$ denote the dislocation core pressure and $F^c$ the local volume change per unit volume induced by the core pressure. When $p_c > 0 \Rightarrow F^c > 0$, the core exerts an outward pressure on the surrounding crystal, so dilatation takes place. Negative $p_c$ would be tensile, causing lattice contraction.

Denoting the volume change per unit reference volume of an element of crystal due to dislocations contained within as $\Delta V/V$, extending prior work [43,45], we find

$$\frac{\Delta V}{V}(\rho, \chi, p_c) = \alpha^e \chi b^2 \rho + \alpha^s (1 - \chi) b^2 \rho + F^c(\rho, \chi, p_c) \tag{16}$$

$$\alpha^e = \frac{\Lambda}{(1-\nu)^2}\left[\frac{1}{2}(1-2\nu)^2(B'-1) + \frac{2}{3}\left(1-\nu+\nu^2\right)\left(G' - \frac{3(1-2\nu)}{2(1+\nu)}\right)\right], \; \alpha^s = \Lambda\left[G' - \frac{3(1-2\nu)}{2(1+\nu)}\right] \tag{17}$$

The first term on the right side of Equation (16) is due to nonlinear elastic or anharmonic effects of edge dislocation components, the second to screw components, and the third to core pressure. Dimensionless factors $\alpha^e$ and $\alpha^s$ contain the combined effects of elastic constants and their pressure derivatives on anharmonicity and dislocation line energy. Parameter $\Lambda$ scales the dislocation energy per unit length [56–58], as discussed following Equation (17) below.

Equations (16) and (17) are derived by making the following substitutions in Equation (74) of ref. [45] for the energies per unit length of screw and edge dislocations, respectively, $E^s$ and $E^e$, and the ratio of ambient shear to bulk modulus, $G/B$:

$$E^s = (1 - \nu)E^e = \Lambda Gb^2, \; \Lambda \approx 1, \; G/B = \frac{3(1 - 2\nu)}{2(1 + \nu)} \tag{18}$$

The explicit effect of core pressure is also newly added in Equation (16), following its possible significance discovered in ref. [46]. As newly derived in Equation (17), the dimensionless volume change factors $\alpha^e$ and $\alpha^s$ depend only on dimensionless constants $\nu$, $B'$, and $G'$. Derivation of Equation (74) of [45] relies on the assumptions that the body is self-equilibrated with a constitutive response described by a hyperelastic energy potential, expanded to order three in the Lagrangian strain (i.e., strain energy with elastic constants of second and third orders). This body contains internal discontinuities associated with jumps in lattice displacements across slip planes, from Burgers vectors of dislocations. Traction is continuous across discontinuity surfaces. The balance of linear momentum and nonlinear elastic constitutive equations are substituted into the equation for vanishing volume-averaged stress. Further assuming isotropic elastic symmetry and algebraic manipulations produces Equation (74) in [45]. This is recast via (18) into (16) and (17) of the current work, to which $F^c$ has been appended.

In Equation (18), the dislocation line energy is approximated as $Gb^2$ for screw dislocations and $Gb^2/(1 - \nu)$ for edge dislocations. These are likely upper bounds among known linear elastic approximations [43,45,56,57], and they omit shielding effects from dislocation structural rearrangements [56–58]. Lower bounds would multiply the predicted dilatation of Equations (14)–(16) by a factor of $\Lambda$ ranging from $1/2$ to $1/(4\pi)$ [43,58].

Listed in Table 1 are elastic coefficients for ten metals used to test the hypothesis framed for the question asked at the beginning of Section 4.3. Isotropic values are for polycrystals from Guinan and Steinberg [59]. Also shown for comparison is the Zener anisotropy factor $A = 2C_{44}/(C_{11} - C_{12})$, with $A = 1$ being isotropic. The larger the departure of $A$ from unity, the less valid the isotropy assumption inherent in Equations (15)–(17). For hcp Mg, elastic anisotropy also depends on other combinations of elastic constants, but anisotropy is generally very low in Mg for all such combinations. Anisotropy itself is also worth considering alone for possible connections to screw core non-planarity and lattice friction.

**Table 1.** Anharmonic volume change factors of dislocations from Equation (15), with elastic properties. Experimental data for $\alpha_{\exp}$ from ref. [35] on Al 1100 and from ref. [34] on Ag, Au, Cu, and Ni.

| Metal | Structure | $B$ [GPa] | $G$ [GPa] | $\nu$ | $B'$ | $G'$ | $A$ | $\alpha^e$ | $\alpha^s$ | $\alpha_{\exp}$ | <111> Spreading |
|---|---|---|---|---|---|---|---|---|---|---|---|
| Fe | bcc | 166 | 82 | 0.29 | 5.3 | 1.8 | 2.37 | 2.13 | 1.31 | - | Yes |
| Mo | bcc | 263 | 125 | 0.29 | 4.4 | 1.5 | 0.72 | 1.66 | 1.02 | - | Yes |
| Ta | bcc | 193 | 69 | 0.34 | 3.2 | 1.1 | 1.56 | 1.14 | 0.74 | - | Yes |
| W | bcc | 310 | 160 | 0.28 | 4.0 | 2.3 | 1.01 | 2.39 | 1.78 | - | Yes |
| Al | fcc | 76 | 26 | 0.35 | 4.4 | 1.8 | 1.22 | 2.14 | 1.46 | 2.04 | No |
| Ag | fcc | 103 | 30 | 0.37 | 6.1 | 1.4 | 3.03 | 1.87 | 1.11 | 1.08 | No |
| Au | fcc | 173 | 28 | 0.42 | 6.3 | 1.1 | 2.88 | 1.61 | 0.94 | 1.08 | No |
| Cu | fcc | 137 | 48 | 0.34 | 5.5 | 1.4 | 3.21 | 1.77 | 1.05 | 1.23–1.68 | No |
| Ni | fcc | 183 | 86 | 0.30 | 6.2 | 1.4 | 2.46 | 1.86 | 0.93 | 1.74–1.78 | No |
| Mg | hcp | 35 | 17 | 0.29 | 3.9 | 1.7 | 0.98 | 1.78 | 1.21 | - | No |

Most notably calculated in Table 1 are values of $\alpha^e$ and $\alpha^s$ that indicate the theoretically predicted importance of anharmonic effects on dilatation from edge and screw dislocations, respectively. For a density of perfect screw dislocations (e.g., <111> screws in a bcc metal of current interest), $f^e = \chi = 0$, and thus $\alpha^e$ and $B'$ would be inconsequential. However, the core structure of a nonplanar, nominally pure screw <111> dislocation contains partial edge



components that sum to zero [5]. Thus, $\alpha^e$ and $B'$ could still be of relevance for correlating non-Schmid effects. As might be expected, results in Table 1 confirm $\alpha^e > \alpha^s$ for all ten metals, meaning dilatation from edge dislocations should exceed that from pure screw dislocations unless core pressure effects are larger from screw dislocations.

Experimentally obtained values of the dimensionless volume change factor, $\alpha_{exp}$, are shown for comparison with model predictions of $\alpha^e$ and $\alpha^s$ in Table 1, where

$$\alpha_{exp} = G\frac{\Delta V/V}{W^c} = \frac{\Delta V/V}{\Lambda b^2 \rho} \qquad (19)$$

Values of $\alpha_{exp}$ are calculated from Equation (19) using experimental data on volume change $\Delta V/V$ and measured stored energy per unit volume of cold work $W^c$ [34] or measured total dislocation line density $\rho$ [35]. Available data do not allow delineation of screw versus edge factors. Equation (18) follows from (15)–(17) with omission of the core term $F^c$, which also cannot be deduced from experimental data. The relatively high value of unity for $\Lambda$ in (17) compensates for the neglect of a distinct core pressure. Agreement between the theory and experiment is respectable since $\alpha^s \leq \alpha_{exp} \leq \alpha^e$ for all fcc metals except Ag. Even for Ag, the lower bound of the theory, $\alpha^s$, exceeds $\alpha_{exp}$ by only 3%. The discrepancy could be attributed to the large anisotropy of Ag (i.e., $A$ exceeding 3), recalling that Equation (17) relies on the isotropic assumption. The theory correctly predicts that Al should have the largest dilatation among fcc metals in Table 1. Excluding anomalous results for Ag, the theory correctly predicts that Au should have the lowest dilatation, with Cu and Ni falling in between, depending on the fractions of edges and screws. Quantitative data were found in the literature only for the five fcc metals in Table 1 and not bcc or hcp. However, Spitzig and Richmond [35] stated that predictions of the analytical theory [51] agreed with experimental measurements of dilatation and dislocation densities in Fe-based metals.

Results in Table 1 show no positive correlations among observed non-Schmid effects [1,5] and any of $\alpha^s$, $\alpha^e$, or $A$. For example, for the two listed Group VIB metals, $\alpha^s$ is larger in W than Mo, but Mo shows stronger non-Schmid effects associated with experimentally measured strength asymmetries [5]. Some of the fcc metals and Mg (hcp) have larger values of $\alpha^s$ than Mo, but these do not demonstrate non-Schmid effects from <111> core spreading. The rank ordering from highest to lowest $\alpha^s$ is W, Al, Fe, Mg, Ag, Cu, Mo, Au, Ni, Ta. Ordering for $\alpha^e$ is similar but not identical: W, Al, Fe, Ag, Ni, Mg, Cu, Mo, Au, Ta.

The present analysis thus leads to the following conclusion: if dilatation from predominantly screw dislocation lines, dislocation core spreading, and non-Schmid effects are positively connected, then the dilatation must be induced from an independent core pressure, rather than anharmonic (i.e., nonlinear elastic) effects alone. A correlation with core pressure appears logical since a larger core pressure might be expected to be exerted by a more disordered non-planar core, leading, in turn, to greater dilatation. Results in Table 1 also show no correlation between elastic anisotropy factor $A$ and non-Schmid effects. For example, the Group VB metal Ta has weaker core spreading than the Group VIB metals W and Mo [5], yet Ta is more anisotropic than W and Mo according to values of $A$.

The maximum magnitude of $\Delta V/V$ from either of the two anharmonic terms in Equation (14) can be estimated as follows. The closest packing of dislocation lines is limited theoretically by the lattice spacing and repulsive forces between atoms to a maximum $\rho \approx 0.01/b^2$, or approximately one dislocation line per square patch of $10 \times 10$ unit cells [60]. Thus, the absolute theoretical maximum dilatation, in percent, from anharmonicity is on the order of $\alpha^s$ or $\alpha^e$. From Table 1 for three bcc metals, this would be a maximum expansion of 0.7% to 2.4%, depending on the particular metal and dislocation character (screw or edge). For the most extreme case of edge dislocations in W, the predicted maximum is 2.4%; the least extreme is for screw dislocations in Ta, predicted at 0.7%. Predictions for Mo fall in between those for W and Ta. If the dislocation line energy is reduced by shielding, a theoretical maximum of around 1% expansion is expected to be more realistic for W. These

theoretical predictions are of comparable magnitudes to those for five fcc metals (validated versus experimental data in Table 1) as well as hcp Mg.

Even in heavily cold-worked metals, the dislocation density is usually several orders of magnitude smaller than the above theoretical maximum, which should limit $\Delta V/V$ accordingly, except in very highly defective local regions at grain boundaries or cell walls. Experiments [35] measured the volume change in Al 1100 as $5 \times 10^{-5}$ at a strain of 0.1 and dislocation density of $3 \times 10^{14}/\mathrm{m}^2$. The theoretical maximum of $\rho$ with a typical Burgers magnitude is around $10^{17}/\mathrm{m}^2$. A dislocation density of $10^{15}/\mathrm{m}^2$ is considered quite large for a typical metal, and this in turn would give a maximum dilatation from anharmonic effects on order of 0.01%, or $10^{-4}$. The pressure differential due to a dilatation from a nucleation of this density of dislocations [61] is around $10^{-4}B$, on the order of 10 MPa for the metals in Table 1. These values of residual volume change and pressure are sufficiently small to omit in the continuum plasticity theory for bcc metals under ordinary conditions (i.e., relatively low strain rates and temperatures), with Peierls yield and flow stresses on the order of 1 GPa, as modeled in Sections 2 and 3 of this work. However, for very soft metals such as Cu, 10 MPa might not be small compared to the yield stress, which can be significantly lower than 100 MPa depending on its purity [62]. The potential significance of plastic volume change under extreme loading conditions involving very high pressures, strain rates, and/or temperatures [62–64] is given further consideration in Appendix A.

## 5. Conclusions

At low temperatures, the yield criterion and plastic potential in bcc metals are decoupled and display a tension–compression asymmetry. The asymmetry is a consequence of screw dislocation non-planarity. Newly proposed in the current work is that the yield stress and plastic potential each include an additional term which quantifies the spreading of the dislocation core structure that leads to non-Schmid effects in plastic flow. The material parameter controlling each new term is interpreted as an atomistically resolved friction coefficient. This coefficient resembles the friction coefficient in Coulomb-type (e.g., brittle or granular) materials, but in bcc metals, the coefficient arises at the atomic scale of the dislocation core. The theoretical analysis in Section 4 justifies the mechanisms-based considerations at the continuum scale.

The theory predicts no obvious correlation between core spreading (which affects the macroscopic friction coefficient) and dilatation from anharmonic effects under isothermal, low-temperature conditions. If a correlation between screw dislocation core spreading and residual lattice expansion exists, as logically hypothesized, such a correlation can be represented through introduction of a finite, repulsive dislocation core pressure. Theoretical predictions also justify the omission of volumetric plastic deformation arising from anharmonic defect fields in bcc metals that show strong non-Schmid effects (i.e., Mo and W) for conventional loading conditions, wherein dislocation densities are not excessive. This conclusion is consistent with theoretical predictions and experimental data on plastic dilatation and stored energy of cold work in fcc metals.

**Author Contributions:** Conceptualization, A.Z. and J.D.C.; formal analysis, A.Z. and J.D.C.; investigation, A.Z. and J.D.C.; writing—original draft preparation, A.Z. and J.D.C.; writing—review and editing, A.Z. and J.D.C. All authors have read and agreed to the published version of the manuscript.

**Funding:** This research received no external funding.

**Data Availability Statement:** Not applicable; manuscript contains no data.

**Conflicts of Interest:** The authors declare no conflict of interest.

## Appendix A. Extension to Dynamic High-Pressure Regimes

The effects of residual lattice dilatation from dislocations on pressure are undeniable in shock compression experiments, as stated in other studies [62–65]. Residual dilatation has been extracted from velocity profile histories in shock compression experiments on

materials of very low flow stress such as pure copper (Cu) [62,65]. Dislocation densities measured in shock-recovered samples after 35-GPa impact exceed $2 \times 10^{15}/\text{m}^2$ [66], and transient densities could be much larger [67]. For example, dislocation densities predicted by discrete dislocation dynamics and atomistic simulations of shockwaves in Cu range from $10^{16}/\text{m}^2$ to $2 \times 10^{17}/\text{m}^2$ for shock pressures from 30 GPa to 75 GPa [67,68]. The volume change from the latter could attain, from Table 1, $0.013\alpha^e \approx 2.3\%$. The pressure change associated with such significant dilatation would be around 3.2 GPa, over 5% of the impact stress [69] and much larger than the yield and flow stresses of Cu [65,68]. In cases such as this, explicit inclusion of plastic volume changes from dislocations in the finite-strain kinematics of continuum crystal plasticity theory [43,60,61,69], or later via Equation (18), is prudent.

In addition, a volume change can be a byproduct of nano-scale dynamic excitations triggered by a collective motion of dislocations [64]. Excitations distort the lattice and, we hypothesize, affect lattice stretch. Under quasi-static conditions, pressure points can be ignored; however, shocks magnify the core pressure and anharmonicity to explicitly contribute to overall dilatant behavior. Dynamic excitations triggered by a synchronized motion of dislocations generate micro-kinetic energy, which in turn may act as phono-to-phonon vibrations; that is, micro-kinetic energy expands the lattice. Dynamic behaviors magnify effects in both time and space. The magnitude of micro-kinetic energy must be large enough for these effects to be noticeable, which happens only under extreme loading conditions. A method has been set forth to calculate micro-kinetic energy [64]; thus, one can introduce additional lattice stretch, a kind of local anharmonic thermal stretch.

The tensor representation concept of Sections 2 and 3 has all the features suitable for a proper physics-based interpretation of these phenomena. The TRM analysis can be easily extended to shock conditions. The second term in Equation (8) provides a means for restraining out-of-plane spreading of dislocations at high pressures. It has been reported that single crystalline Ta, among other metals, subjected to impact loading, experiences highly localized plastic slip [70]. Dynamic loading alters the mechanisms of plastic flow. That brings us back to Equations (7) and (9), where we search for explanations of the intriguing problem. The equivalent stress $\sigma_{eq}$ in (9) consists of shear stress along the glide plane and includes the non-glide contribution. One may argue that, because it tightens interatomic spaces, high shock pressure also over-constrains spreading of dislocation cores. For this reason, we modify Equation (7) by rendering the out-of-plane contribution the needed sensitivity to the changes in mass density:

$$M_{ij} = \alpha \left[ \left( N_{ij}^1 - N_{ij}^3 \right) + \mu_\varphi \left( \delta_{ij}/3 - \frac{\rho_0}{\rho} N_{ij}^2 \right) \right] \tag{A1}$$

Here, $\rho_0$ and $\rho$ are initial and current mass densities, not dislocation densities. In a linear elastic regime, the ratio $\rho_0/\rho$ is nearly unity, so its effects can be omitted. At high pressure, equations of state quantify pressure-volume-temperature-entropy responses [62–64].

Suppose that the pressure is 50 GPa, which translates to $\rho_0/\rho \cong 0.8$. Now, temperature is high due to adiabatic shock heating and thermoelastic coupling, and, therefore, we conclude that $\mu_0 = 0$. This also means that the associated flow rules are reestablished. In Figure A1, the yield surface marked in black is unaffected by the change of mass density ($\rho_0/\rho = 1$). The second envelope, marked in red, is plotted for $\rho_0/\rho \cong 0.8$. Shapes of the envelopes are distinctly different.

In the next step, we assume a plate impact problem where the uniaxial stress points in direction 1. We calculate the plastic strain rates along three directions, mainly $r_{12} = M_{11}/M_{22}$, $r_{13} = M_{11}/M_{33}$, and $r_{23} = M_{22}/M_{33}$. The ratios are $r_{12} = -2.172$, $r_{13} = -2.172$, and $r_{23} = 1$, respectively. We find that there is a small increase of volume $\dot{\varepsilon}_{kk}^p = 0.0367 \, \dot{e}_{eq}^p$ since the tensor in (7) is no longer traceless. Thus, a high-pressure loading generates small plastic dilatation. Experimentally observed perturbations in surface velocity profiles for shock compression of W- and Al-based metals [71,72] suggest a pressure variation, which we argue can be a result of residual dilatation from magnified core pressure and/or

anharmonic effects, especially since W and Al have the largest values of dilatation factors $\alpha^e$ and $\alpha^s$ in Table 1. However, this source of perturbations cannot, in general, be uniquely separated from other phenomena related to microstructure heterogeneities.

Shock relief ($\rho_0/\rho = 1.2$) generates external tensile pressure, and the trends are reversed. Now, the ratios are smaller, $r_{12} = -1.854$, $r_{13} = -1.854$, and $r_{23} = 1$, and the rate of plastic dilatation $\dot{\varepsilon}^p_{kk} = -0.0367\,\dot{e}^p_{eq}$ opposes the elastic stretch. Thus, in tension, the plastic contraction of dislocation cores tends to absorb the already large interatomic distances. A local volume reduction can also be associated with a transient decrease in dislocation density after the shockwave has passed and the material relaxes to equilibrium.

Incompressibility is the commonly used assumption in classical constitutive models [73]. The phenomenology omits non-planarity of plastic flow and thus cannot predict phenomena associated with plastic volume changes. In the proposed continuum description, the second term in Equation (7) includes the local mass density ratio, which enables a richer description of the metal behavior at shock conditions. In conclusion, the proposed concepts can be used to study the plastic responses of bcc metals at low temperatures, as we have demonstrated for Mo in Section 3, and can be extended to extreme high-pressure conditions, as newly proposed in this appendix.

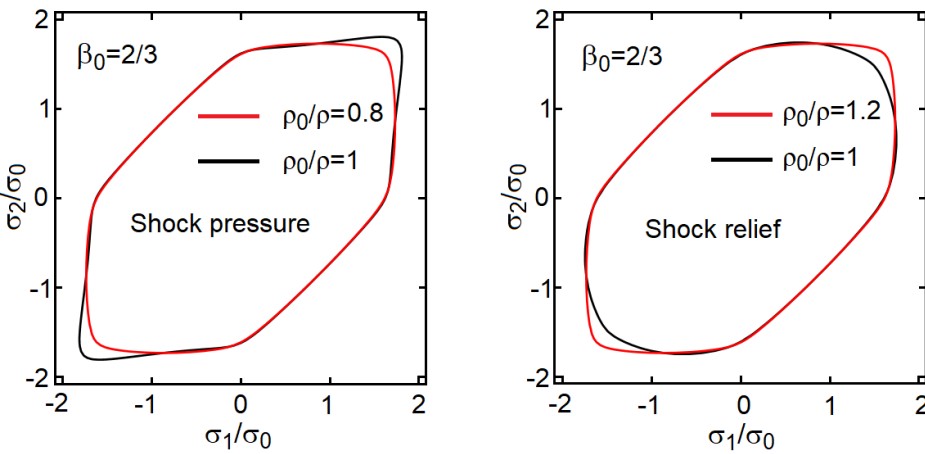

**Figure A1.** Yield surfaces are plotted for a non-shocked metal (black lines) and for the same metal at high shock pressure and the subsequent pressure relief (red line).

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
