# Peer review of "Yield Surfaces and Plastic Potentials for Metals, with Analysis of Plastic Dilatation and Strength Asymmetry in BCC Crystals"

_metals, doi:10.3390/met13030523_

Round 1

Reviewer 1 Report

The authors present an extension to existing plastic flow theory in which frictional drag coefficients are adopted to enable non-work-conjugate flow (i.e. decoupled plastic work and yield stress evolution) representation in a continuum constitutive law. This seems to me like a somewhat obscure but potentially interesting development. However, the manuscript does a poor job placing within the proper context and highlighting the novelty while demonstrating the validity of the method against experimental observation. I and thus recommending a major revision: although the fundamentals seem mostly alright (although missing experimental validation), the ms itself would benefit from substantial reconsideration. I have the following specific recommendations:

1.      The title is overly generic, and does not describe the work to potential readers. Please revise.

2.      Typos/typesetting to fix, e.g., line 79, 108. A further careful edit is suggested.

3.      This is seemingly a substantial extra commentary/descriptions, some of which seem overly hypothetical or speculative (e.g. paragraph on lines 309-321), with many references to further “in-depth” or “quantitative” studies being required. One begins to wonder what value this work has, if so much requires future study. Really, a lot of sections 2, 3 and 4 could be removed to focus on the new developments, if there are any.

4.      Over all, the new developments seem rather minor (i.e., addition of one term to existing yield stress and plastic potential equations) – most equations and discussion relies on references rather than new derivations, it seems. The authors should separate background and prior works from the current developments. Section 4 seems to suffer from this in particular – I’m not really sure what section 4 is doing here.

5.      While the authors claim an atomistic basis for the proposed work, mathematically this is simply a minor tweak to existing empirical equations (albeit with theoretical inspiration). To my reading, which may be incomplete, the addition of new so-called friction coefficients is simply a mathematical expedient to represent non-planar response – since this is a macroscopic continuum model, no information regarding atomic-scale interactions has been retained fundamentally within the model, just a somewhat hand-waving behavior. While fundamentally appropriate, the justification should not be so forced. It seems to be an empirical model that matches some type of simulated material response, pages of description regarding known dislocation mechanics are not needed and detract from the novelty.

6.      The practical application of the current work and validation against ground-truth experimental evidence is lacking, despite the extensive descriptions.

7.      The authors would do well to present the full derivations of their new model, ideally from the most basic equations step-by-step to the final form, especially where such derivations are new. In some cases this may be most appropriate in an appendix, but they should be shown somewhere for the interested reader to follow.

Author Response

General comment

We would like to thank the Reviewer. The comments, criticism and suggestions are well-pointed, and we made an effort to address all the issues.  

Reviewer 1

The authors present an extension to existing plastic flow theory in which frictional drag coefficients are adopted to enable non-work-conjugate flow (i.e. decoupled plastic work and yield stress evolution) representation in a continuum constitutive law. This seems to me like a somewhat obscure but potentially interesting development. However, the manuscript does a poor job placing within the proper context and highlighting the novelty while demonstrating the validity of the method against experimental observation. I and thus recommending a major revision: although the fundamentals seem mostly alright (although missing experimental validation), the ms itself would benefit from substantial reconsideration.

We acknowledge the criticism and decided to rewrite the paper, have included additional explanations and provided physics-based reasoning. Please note the changes marked in green.

I have the following specific recommendations:

  1. The title is overly generic, and does not describe the work to potential readers. Please revise. See later part.

Title has been revised.

  1. Typos/typesetting to fix, e.g., line 79, 108. A further careful edit is suggested.

These have been fixed along with a few others discovered in proofreading.

  1. This is seemingly a substantial extra commentary/descriptions, some of which seem overly hypothetical or speculative (e.g. paragraph on lines 309-321), with many references to further “in-depth” or “quantitative” studies being required. One begins to wonder what value this work has, if so much requires future study. Really, a lot of sections 2, 3 and 4 could be removed to focus on the new developments, if there are any.

It might be worth stating that the formal structure of the flow tensor M and the theoretical dislocation-based analysis inspired us to find physics-based explanations. Please note that the mathematical structure of the tensor M was derived, and not assumed, where the von Mises flow mechanism established the departure point for the analysis to follow. In sections 2 and 3, we describe the methodology in detail. In the approach, generic mechanisms of plastic flow are studied first, appropriate stimuli such as stress are determined, and later the two are connected through the tensor representation method. In this manner, the try-and-error search for empirical mathematical description has been greatly reduced. The model for molybdenum seems to be a good example for the above.

The speculative comments and calls for future work have been reduced. The hypothetical discussion in original lines 309-321 has been improved to include more physics justification and supporting calculations in Appendix A for shock loading. Content of Appendix A is also novel, but we placed it in an Appendix to avoid confusion with the main parts of the paper that are focused on low-temperature, less extreme loading conditions.

  1. Over all, the new developments seem rather minor (i.e., addition of one term to existing yield stress and plastic potential equations) – most equations and discussion relies on references rather than new derivations, it seems. The authors should separate background and prior works from the current developments. Section 4 seems to suffer from this in particular – I’m not really sure what section 4 is doing here.

The purpose of Section 4 is now clearly defined in the first two paragraphs, lines 243—259 of pp. 6-7. Section 4 has been separated into 3 sub-parts. Sections 4.1 and 4.2 contain background information for readers less familiar with the subject and on prior theory upon which new analysis of Section 4.3 is based. Section 4.3 contains the (relatively few) new derivations, tabulated results, and new deductions drawn from these new results for some bcc metals.

  1. While the authors claim an atomistic basis for the proposed work, mathematically this is simply a minor tweak to existing empirical equations (albeit with theoretical inspiration). To my reading, which may be incomplete, the addition of new so-called friction coefficients is simply a mathematical expedient to represent non-planar response – since this is a macroscopic continuum model, no information regarding atomic-scale interactions has been retained fundamentally within the model, just a somewhat hand-waving behavior. While fundamentally appropriate, the justification should not be so forced. It seems to be an empirical model that matches some type of simulated material response, pages of description regarding known dislocation mechanics are not needed and detract from the novelty.

Once again, please note that the macroscopic continuum model is based on a generic description of the plastic flow (eq. 7). The continuum-level formalism takes over when applying the tensor representation method, where the connection between the properties of M and stress is established. The resultant yield stress and plastic potential are a direct consequence of the above. Thus, the model goes much beyond the classic empiricism. It is true that the tensor representation method was introduced sometimes ago but has not been popularized in the modeling community as it should be. TRM is a truly powerful tool in the hands of a modeler. We provided additional details about the method and included intermediate derivations. In overall, we believe the tie-ins to dislocation mechanics offer an insightful connection among model origins, parameters and the physics, and thus should be maintained.  The overall length of the paper at 16 pages is not excessive.

  1. The practical application of the current work and validation against ground-truth experimental evidence is lacking, despite the extensive descriptions.

Regarding yield and flow potentials, please note a new discussion on lines 230-248 on pp. 6-7, with added references to experiments [30-32].

Regarding plastic dilatation from dislocations, experimental validation of the theoretical calculations (eqs. 14-16) has been added for five fcc metals. Results are shown now in Table 1 and are discussed on p. 10, lines 416-435. No experimental values could be found in the literature for the bcc metals or for hcp Mg.

  1. The authors would do well to present the full derivations of their new model, ideally from the most basic equations step-by-step to the final form, especially where such derivations are new. In some cases this may be most appropriate in an appendix, but they should be shown somewhere for the interested reader to follow.

After reading the comment, we acknowledged the shortcoming in the original description and have substantially rewrote sections 2 and 3 of the paper. In addition, we included a novel part which extends the model into shock conditions. 

Furthermore, the new derivations of Section 4.3 have been highlighted in lines 374-391 of p. 9, including addition of prior missing intermediate steps in eq. (16).  A rigorous description of all steps leading up to eq. (74) of ref. [45] from “first principles” requires 70+ equations (e.g., 20+ pages in Chapter 7 of the book ref. [43]), and can also be found in ref. [45], to which the reader has been referred. A reprint of reference [45] is available for free public download, as now annotated in the citation entry on p. 15. Reproduction of the entire derivation, which contains heavy index notation for finite deformations and nonlinear third-order elasticity, would overwhelm the current paper.

Reviewer 2 Report

The manuscript contains analysis of yielding and plastic flow of a BCC material with focus on Mo, under low temperature condition. A large part of the manuscirpt is devoted to discussion of screw dislocations in bcc metals, core spreading, dilatation, anharmonic effects. Only a smaller part of the paper deals with the extension of Von Mises criterion, to include frictional effect by atomistically-resolved friction coefficient in order to model yield asymmetry. Hence, the title: "Yield Surfaces and Plastic Potentials for Metals" is very misleading and do not correctly reflect the content of the manuscript. 

The authors apply Von Mises and Tresca criterion to model a BCC material, without referring to the classical way to model plasticity of a BCC material by Hersey-HOsford yield surface with high exponent of about 6, as a result from the Bishop-Taylor-Hill analysis. In addition, some relevant references on approaching the yield asymmetry are missing, as: Orthotropic yield criterion for hexagonal closed packed metals, IJP, or  Orthotropic yield criteria for description of the anisotropy in tension and compression of sheet metals, IJP. 

Newertheless, use of Von Mises criterion should be justified as there are more advanced criterions developed. 

In the Figure 1, it is surprising to see that introduction of a friction parameter results in yield criterion which is similar to a Hersey-Hosford one with a quite large exponent, perhaps a 20. It is not obvious to realize this.  Hence, more explanation of the role of beta0 would be wlecome. When discussed FCC material, neither Von Mises nor Tresca are the best continuum representations, but it is Hersy-Hosford with exponent 8. This should be mentioned.

What exactly represents angle phi? The eigentensors N^m_ij are dependent on the angle phi but then it gives that also principal stresses depend on \phi, as \sigma_1 = N^1_ij \sigma_ij. If N^1_ij depends on \phi, then \sigma_1 also does?

Line 79 – “The In the first step” contains a typo

Line 109 – fix equation numbering

Line 336 – remove dots for multiplication of scalars

Author Response

General comment

We would like to thank the Reviewer for well-pointed comments and suggestions. We made an effort to address all the issues.   

Reviewer 2

The manuscript contains analysis of yielding and plastic flow of a BCC material with focus on Mo, under low temperature condition. A large part of the manuscript is devoted to discussion of screw dislocations in bcc metals, core spreading, dilatation, anharmonic effects. Only a smaller part of the paper deals with the extension of Von Mises criterion, to include frictional effect by atomistically-resolved friction coefficient in order to model yield asymmetry. Hence, the title: "Yield Surfaces and Plastic Potentials for Metals" is very misleading and do not correctly reflect the content of the manuscript.

Title has been revised.

Also, the part that is based on the tensor representation method has been substantially extended. We included additional derivations and the model justification. The main point here is that any generic flow mechanism can now be coupled with the driving forces (tensors). Thus, when the two are understood, phenomenology or empiricism of the mathematical description is greatly reduced. In this case, the formal structure of the flow mechanism (eq. 7)  was inspired by the von Mises flow and then the mechanism was further generalized.  

The authors apply Von Mises and Tresca criterion to model a BCC material, without referring to the classical way to model plasticity of a BCC material by Hershey-Hosford yield surface with high exponent of about 6, as a result from the Bishop-Taylor-Hill analysis.

Thank you for the remainder. These seminal works have now been noted on p. 2, lines 61-67. Citations to the refs. [14-18] have been added.

In addition, some relevant references on approaching the yield asymmetry are missing, as: Orthotropic yield criterion for hexagonal closed packed metals, IJP, or Orthotropic yield criteria for description of the anisotropy in tension and compression of sheet metals, IJP.

The requested citations, as well as the anisotropic Hill criterion, have been added on p. 2, lines 67-69, refs. [19-21].

Nevertheless, use of Von Mises criterion should be justified as there are more advanced criterions developed.

Please note that we do not use the von Mises criterion. We have shown that the criterion becomes a special case which is not exercised in our paper.

In the Figure 1, it is surprising to see that introduction of a friction parameter results in yield criterion which is similar to a Hersey-Hosford one with a quite large exponent, perhaps a 20. It is not obvious to realize this.  Hence, more explanation of the role of beta0 would be welcome. When discussed FCC material, neither Von Mises nor Tresca are the best continuum representations, but it is Hersy-Hosford with exponent 8. This should be mentioned.

Thank you for the comment. We included additional explanations in lines 203-213. In simple terms, the parameter explicitly quantifies the non-Schmidt factor. As explained on p. 4, lines 158-169, beta_0 is proportional to the friction coefficient mu_\phi, which accounts physically for enhanced glide resistance from dislocation core spreading in bcc metals at low temperatures.

The Hershey-Hosford [14] approach has been cited on p. 4, lines 174-177, as requested in the context of Fig. 1.

What exactly represents angle phi? The eigentensors N^m_ij are dependent on the angle phi but then it gives that also principal stresses depend on \phi, as \sigma_1 = N^1_ij \sigma_ij. If N^1_ij depends on \phi, then \sigma_1 also does?

In frictional materials, the angle has a clear geometric interpretation and quantified roughness of the shear plane, where phi is the angle of asperities. In here, the angle evaluates the non-planarity of plastic flow and that, in essence, has a similar interpretation. In metals though, the angle estimates the flow deviation from the dominant plane of slip. Additional comments are included in lines 158-161 and 203-207.

Line 79 – “The In the first step” contains a typo

Corrected.

Line 109 – fix equation numbering

Corrected.

Line 336 – remove dots for multiplication of scalars

Corrected.

Reviewer 3 Report

In the present work, the authors analysed the yield criterion and plastic potential in bcc metals. I have carefully read and considered your work. The research topic is relevant. The results obtained are of interest. The text of the paper is clear and easy to read.  The conclusions of the paper are consistent with the presented evidence and arguments.

Overall, the idea and methodology presented in the manuscript are sound, and the results are interesting. However, there are a few areas that could be improved to enhance the clarity and impact of the work.

The title is too general if this work is focused only on the low temperature responses of body-centered-cubic (bcc) metals.

Authors are encouraged to be included in the references English version of the fundamental paper [9]: https://am.ippt.pan.pl/am/article/viewFile/v56p173/pdf

line 79: There is stylistic error.

Author Response

General comment

We would like to thank the Reviewer for an-in-depth analysis of our work. The comments and suggestions helped us to improve the manuscript and for that we are grateful. We made an effort to address all the issues.   

Reviewer 3

In the present work, the authors analysed the yield criterion and plastic potential in bcc metals. I have carefully read and considered your work. The research topic is relevant. The results obtained are of interest. The text of the paper is clear and easy to read.  The conclusions of the paper are consistent with the presented evidence and arguments.

Overall, the idea and methodology presented in the manuscript are sound, and the results are interesting. However, there are a few areas that could be improved to enhance the clarity and impact of the work.

Thank you for the comments. We made additional improvements, included explanations, added additional section (Appendix), and focused on bridging the continuum level analysis and the dislocation-based considerations.

The title is too general if this work is focused only on the low temperature responses of body-centered-cubic (bcc) metals.

Title has been revised.

Authors are encouraged to be included in the references English version of the fundamental paper [9]: https://am.ippt.pan.pl/am/article/viewFile/v56p173/pdf

This reference is now included as ref. [10] in the list of cited works.

line 79: There is stylistic error.

Corrected.

Round 2

Reviewer 2 Report

The authors have implemented suggested changes well. It is very appreciated that the chapter on tensor representation has been extended, which makes it easier for a reader to follow the main idea. I see that the text contains many short equations, that are important ingrediences of the model, in lines.  At this point, I suggest a minor improvement and to make in-line equation on the line 154 on \mu_0 linked to the beta_0 n a self-standing equation number 8, as it is an important ingredience to the whole model. I suggest to do the same in the Section 3, so that the reader can pick the equations and build the model , instead of finding the relevant and important links in the text.

Author Response

Dear Reviewer,

Thank you for your previous comments and current suggestions. The two equations are self-standing now (Eqs. 8 and 12). 

Best regards,

Alek and John